# Integrating human behavior and snake ecology with agent-based models to predict snakebite in high risk landscapes

Eyal Goldstein[1]*, Joseph J. Erinjery[1,2], Gerardo Martin[3,4], Anuradhani Kasturiratne[5], Dileepa Senajith Ediriweera[6], Hithanadura Janaka de Silva[7], Peter Diggle[8,9], David Griffith Lalloo[10], Kris A. Murray[3,4,11], Takuya Iwamura[1,12]*

1 School of Zoology, Department of Life Sciences, Tel Aviv University, Tel Aviv, Israel, 2 Department of Zoology, Kannur University, Kannur, India, 3 MRC Centre for Global Infectious Disease Analysis, Department of Infectious Disease Epidemiology, School of Public Health, Imperial College London, London, United Kingdom, 4 Grantham Institute—Climate Change and Environment, Imperial College London, London, United Kingdom, 5 Department of Public Health, Faculty of Medicine, University of Kelaniya, Kelaniya, Sri Lanka, 6 Health Data Science Unit, Faculty of Medicine, University of Kelaniya, Kelaniya, Sri Lanka, 7 Deparment of Medicine, Faculty of Medicine, University of Kelaniya, Ragama, Sri Lanka, 8 CHICAS, Lancaster University Medical School, Lancaster, United Kingdom, 9 Johns Hopkins Bloomberg School of Public Health, Baltimore, Maryland, United States of America, 10 Liverpool School of Tropical Medicine, Liverpool, United Kingdom, 11 MRC Unit the Gambia at London School of Hygiene and Tropical Medicine, Atlantic boulevard, Fajara, The Gambia, 12 Department of Forest Ecosystems and Society, College of Forestry, Oregon State University, Corvallis, Oregon, United States of America

* pogoyoly@gmail.com (EG); takuya.iwamura@oregonstate.edu (TI)

**Data Availability Statement:** All relevant data are within the manuscript and its Supporting Information files.

## Abstract

Snakebite causes more than 1.8 million envenoming cases annually and is a major cause of death in the tropics especially for poor farmers. While both social and ecological factors influence the chance encounter between snakes and people, the spatio-temporal processes underlying snakebites remain poorly explored. Previous research has focused on statistical correlates between snakebites and ecological, sociological, or environmental factors, but the human and snake behavioral patterns that drive the spatio-temporal process have not yet been integrated into a single model. Here we use a bottom-up simulation approach using agent-based modelling (ABM) parameterized with datasets from Sri Lanka, a snakebite hotspot, to characterise the mechanisms of snakebite and identify risk factors. Spatio-temporal dynamics of snakebite risks are examined through the model incorporating six snake species and three farmer types (rice, tea, and rubber). We find that snakebites are mainly climatically driven, but the risks also depend on farmer types due to working schedules as well as species present in landscapes. Snake species are differentiated by both distribution and by habitat preference, and farmers are differentiated by working patterns that are climatically driven, and the combination of these factors leads to unique encounter rates for different landcover types as well as locations. Validation using epidemiological studies demonstrated that our model can explain observed patterns, including temporal patterns of snakebite incidence, and relative contribution of bites by each snake species. Our predictions can be used to generate hypotheses and

**Funding:** This work was supported by the Medical Research Council [MP/P024513/1]. The grant was obtain by KM, TI, DGL, HJdeS, and PJD. The funders had no role in study design, data collection, and analysis, decision to publish, or preparation of the manuscript.

**Competing interests:** The authors have declared that no competing interests exist

inform future studies and decision makers. Additionally, our model is transferable to other locations with high snakebite burden as well.

## Author summary

Snakebite is a neglected tropical disease affecting millions, and a major cause of death of agricultural workers in the tropics. In this research, the authors have developed a simulation model that includes data for agricultural activity across the days and seasons, as well as snake behavioral patterns, and the times and locations humans and snakes meet. Using this model, they predicted observed seasonal snakebite patterns in Sri Lanka, and they successfully showed how these patterns vary between different agricultural activities, including seasonal rice cultivation, and rubber and tea harvests. The findings arising from this study demonstrate that different combinations of human and snake activity, including species and farming practice differences, are likely to generate differences in snakebite patterns across locations. This model could be applied to analyze and predict snakebite in tropical regions around the globe to help mitigate the problem.

## Introduction

Globally, five million people are bitten by snakes every year, resulting in approximately 94,000 deaths out of 1.8 million envenoming cases, and up to 400,000 morbidities [1,2]. Most of this burden occurs in the tropics of south east Asia and Sub Saharan Africa [2]. Despite its impacts, snakebite is still considered a neglected tropical disease that is concentrated among the poorest of the poor [2,3], and this may have contributed to the lack of funding and scientific research on snakebite relative to other disease of similar or lesser burden [3–5]. In 2017 snakebite was formally identified a neglected tropical disease by the World Health Organization [3], which prompted the scientific community to increase efforts for combating this disease, including the development of a global snake bite strategy and roadmap [6].

Several past studies have hypothesized on the importance of overlap between snake and human activities as a cause of snakebite patterns (e.g. [7–9]). However, previous research on snakebite has relied heavily on correlative models, that statistically relate bite data (e.g., from hospital admissions) to a range of social and, less often, environmental variables to identify key risk factors [10]. Such studies include those which incorporate climatic factors such as precipitation, humidity, and mean temperature [4,11–13], social factors including human population density, poverty, and farming activities [4,12,14–17], and ecological factors such as snake activity or distribution information [11,14,18,19]. For example, Yañez-arenas et al., (2016) [19] show a correlation between snake distributions and bites, and Akani et al., (2013) [14] matched patterns of snake activity with agricultural activity of local farmers across different months to reveal correlation with snakebite occurrences. However, no studies have yet taken a mechanistic socio-ecological approach that integrates both human and snake distributions and behaviors to investigate the ways in which snakebite epidemiology is simultaneously shaped by ecology, climate, and landscape characteristics.

Agent based modelling (ABM) is a bottom up approach for modeling complex and adaptive systems. ABMs are comprised of collections of individuals (agents) that are programmed to display behavioral traits, while their interactions with each other generate phenomena at a higher level [20–23]. ABM is used both for representing the internal dynamics of complex

systems, and discovering emergent patterns that may be found in those systems [24,25]. Spatially explicit social-ecological dynamics are increasingly modelled using an ABM approach (e.g.: [22,26,27]), such as those involving land use and land cover change [26,28,29]. ABM has also been used for modelling ecological epidemiology, including zoonotic disease transmission across landscapes (e.g.: [30]), mosquito behavior in models for malaria transmission [31], rabies transmission among foxes [32], and the spread of foot and mouth disease [33]. With snakebite sharing many socio-ecological characteristics with zoonotic diseases [10], ABM is an ideal and novel approach to investigate the epidemiology of snakebite from a mechanistic perspective (see Fig 1).

Sri Lanka is a global snakebite hotspot [2]. It has been estimated that nationally there are more than 80,000 snakebites a year, 30,000 of which involve envenoming. Due to high quality health systems, only around 400 of these result in deaths annually [12]. Nevertheless, morbidity is considerable and the total annual economic burden on households of snakebite envenoming in Sri Lanka amounts to almost $4 million, while it costs the public health system around $10 million per year [34]. Sri Lanka is home to over a hundred snake species, as well as nine medically important land snake species, including: *Daboia russelii*, *Naja naja*, *Bungarus caeruleus*, *Bungarus ceylonicus*, *Echis carinatus*, *Hypnale hypnale*, [35]. Many of these species also contribute to an extensive burden in neighboring regions in South Asia [36]. Previous studies have shown that the frequency of snakebites in Sri Lanka is spatially correlated with climatic, geographic, and socio-economic factors, such as ethnicity, age, occupation, and income [12], with bites occurring seasonally (primarily in the months of November-December, March-May and August) [7]. Snakebite incidence is broadly congruent with the geographical patterns of snake species occurrence across the island [37].

In this study, we integrate socio-ecological factors associated with snakebites in Sri Lanka into a single model by constructing an ABM simulation based on detailed datasets of snake

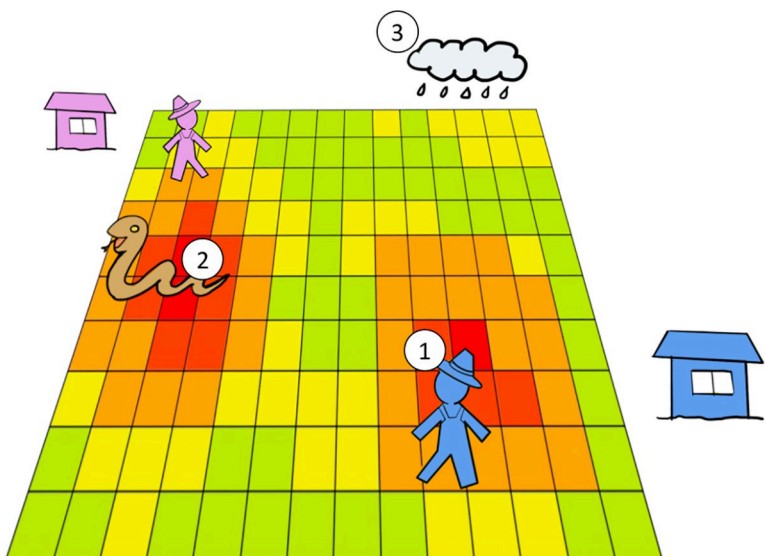

**Fig 1. Modeling approach: Our model simulates daily and seasonal cycles.** A day is represented as 24 time steps. **1. Farmer agent**: Farmer agent has its own daily/seasonal activity schedule according to farmer types (rice, tea, rubber). It owns its piece of crop land. Farmer agent commutes from its home location to its field. It moves inside of her crop area. **2. Snake activity layer:** Snake activity level is determined by the snake species, crop types (habitat types) and precipitation. Snake species determines its distribution probability, habitat preferences, daily/seasonal activity schedule and attack rate. **3. Precipitation cycle**: Precipitation affects snake activity and farmer's activity.

distributions, snake behaviors, landscape characteristics, and farmers' behavioral patterns. Sri Lanka provides an ideal case study for modeling snakebite mechanisms in this way, as not only is it a global hotspot of snakebite, it also provides highly reliable snakebite incidence data and has a high volume of accumulated medical research from which the model can be developed and validated [7,12,35,37]. We developed a spatially explicit ABM to analyze the spatio-temporal overlap between the different medically important snake species and farmers of different crops in Sri Lanka, and integrated climate and landcover as drivers of human-snake interaction across different affected landscapes, in order to create a predictive model that can inform both future research as well as decision makers.

## Materials and methods

### Ethics statement

Our research has been reviewed by the ethics review committee of the faculty of medicine, University of Kelaniya, Kelaniya, Sri Lanka 11600, reference number p/22512/2018. Our study included permission of written consent by all participants who were interviewed during the field work.

**Agent based modeling.** Agent based modelling (ABM) is a bottom up approach for modeling complex and adaptive systems using autonomous agents, that explain macro level phenomenon [20–23]. ABM is used both for studying complexity that is not easily reducible to differential equations, and discovering emergent patterns and phenomenon found in those systems, as well as study the internal dynamics of these system [24,25]. ABM has been extensively used in different fields of study for modeling complex phenomenon, such as social, political, and economical science [24]. There are now multiple programs used for ABM, including NetLogo [38], Repast [39], as well as the SpaDES package in *R* [40]. Recently, spatially explicit social-ecological dynamics are increasingly modelled using an Agent-based modeling approach (e.g.:[22,26,27,41]). It is commonly used for modelling social behavior including modeling land use and land cover change [26,28,29], as well as zoonotic disease transmission across landscapes (e.g.: [30]).

We used Netlogo [38] to develop a spatially explicit model that represents the dynamics of snakebites among farmers (S1 Fig). The model simulates real landscapes in the Study Area, each of which is represented by a 2x2 km study location comprised of a matrix of 10x10m grid cells. We simulated 17 study locations in total.

For the design and analysis of our model we used pattern oriented modelling (POM) [42,43]. This approach emphasizes use of multiple patterns at different hierarchical scales for calibration and validation in order to reduce uncertainty in model structure and parameters. This approach allows us to examine not only large scale phenomena (such as macro level epidemiological observations), but also probe the dynamics and intricacies of the mechanism(s) that may be hidden or unobservable by just examining the different patterns individually.

The pattern oriented modelling protocol is comprised of four steps [42]: 1) aggregate known biological data regarding a system and use it to construct a model that is related to a hypothesis and is theoretically capable of reproducing previously observed patterns; 2) determine the parameter values of the system; 3) compare systematically between the independently observed data and those patterns predicted by the model, which may involve iteratively improving the model by removing certain parameters or choosing combinations of parameters that are more plausible or better represent observed patterns; and 4) look for secondary predictions in the model, which are different from the original patterns to which the model was compared during the third step of the process.

For each one of the locations studied, the model uses a range of input data to simulate the movement and interactions of different 'agents' among cells for a fixed duration. We used a discrete time series comprised of both months and hours. Each month is condensed to 24 timesteps which are representative of individual hours of the daytime, and the simulation is performed across the 12 months of the year, comprising of 288 timesteps in total. Parameters and variables in the simulation are recorded and updated both hourly and monthly, depending on the agent (snake seasonal activity and farmers' working schedules update at the beginning of each month; snake daily activity is updated at the beginning of each hour).

There are two types of agents in the model: farmers and snakes. Farmers are able to work in multiple land cover classes, depending on seasonal needs (see 'Recording Farmer Characteristics' below). Farmers have a state variable of working schedule, which includes the land cover type they should be farming, time of day they begin to work, and the number of hours they will spend working in that land cover class. Using the work schedule, the farmers move between the land cover they are farming and their home.

Each snake agent is characterized by a set of ecological and behavioral traits, including: species, daily activity, habitat preference, aggressiveness, and seasonal activeness. Each species is given a set of probabilities for movement between land cover classes depending on the land association factor (see "Snake distribution and behaviour" below) and number of patches for each land cover class (see "Remote sensing dataset" below).

The influence of the environment on agent activity is represented by climatic variables (precipitation and number of non-rainy days (see "Climate dataset" below)).

**Study area and spatial data.** We focused our modelling effort on the district of Ratnapura in the wet zone of Sri Lanka, which is characterized by high precipitation (see Fig 2). This district has a great diversity of crop types, including tea, rubber, coconut, as well as rice cultivation albeit practiced here on a smaller scale in comparison to other zones of Sri Lanka due to topographic conditions [44–46]. Within the district, we focused our research on four different divisions (Eheliyagoda, Balangoda, Kalawana, and Embilipitiya) that represented the variation in crop types within the district, and at each division level we ran simulations on between 4–5 locations, with 17 locations in total (see Fig 3).

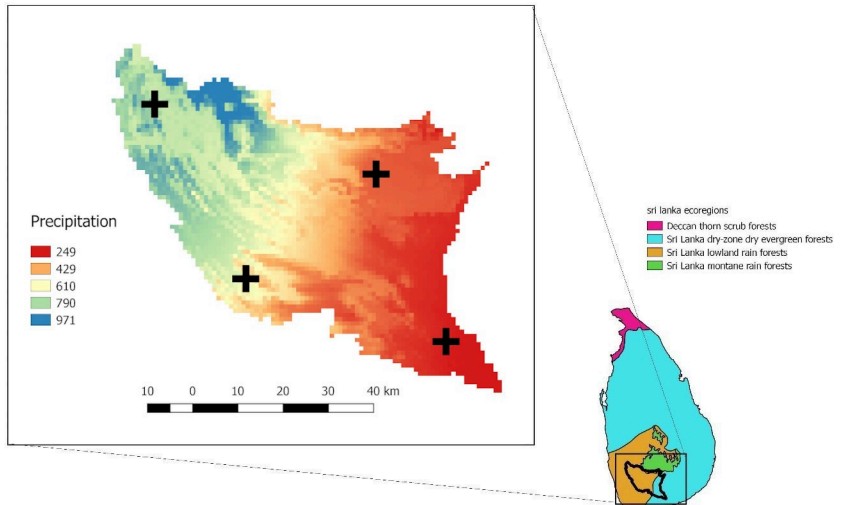

**Fig 2. Ecoregions of Sri Lanka [47].** Annual precipitation of the Ratnapura district (Bioclim variable 12; [48]). The four different divisions (Eheliyagoda–northwest, Kalawana–southwest, Balangoda–northeast, and Embilipitiya—southeast) used in analyses are marked. The Ratnapura district borders on the highlands in the center of the country, the dry zone in the south east, and is part of the wet zone in its center and west.

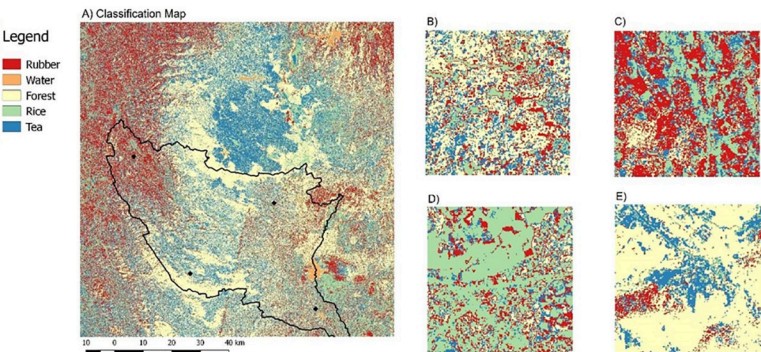

**Fig 3. Classification map using support vector machine (SVM).** A) Classification map for the Ratnapura region created using a SVM classification and sentinel 2 satellite imagery. The Ratnapura district border is marked on the map (black line), and the four divisions where we conducted field work and ran simulations are marked (black dots, see also Fig 2). Variation in landcover types can be observed between locations, with B) the north east (Balangoda) having a mixture of all landcover types, C) the north west (Eheliyagoda) containing a high concentration of rubber plantations, D) the south east (Embilipitiya) containing a high concentration of rice farming, and E) the south west (Kalawana) containing many tea plantations next to forests.

*Landcover* - The main attribute of each cell in the model is its landcover type (Rice, Tea, Forest, Rubber, Home). We used Sentinel-2 remotely sensed images from 2017 to produce vegetation type classification maps (Tile T44NMN and relative orbit numbers R119 & R076), which were chosen based on quality of images and percentage of cloud cover. Tiles were downloaded from the USGS earth explorer portal and were processed using the SNAP program and the Sen2cor plugin [49]. After removing cloud cover, the tiles were merged into a single tile before classification.

We classified the images into five different landcover types giving importance to major crop types and vegetation in the district: forest, rubber, tea, paddy, and water bodies, with a resolution of 10 x 10m (Fig 3). The classification was made using two different supervised classification algorithms: support vector machine (SVM) and maximum likelihood (ML), with 100 training polygons for each land cover type. We used spectra from 4 different bands and NDVI index for classification (band number 2 –Blue, band number 3 –green, band number 4 –red, and band number 8 –near-infra red), with band numbers 4 and 8 used for calculation of the NDVI index. We obtained an overall accuracy of 83.2% and kappa coefficient of 0.68 for the SVM classification and an accuracy rate of 80.7% and kappa coefficient of 0.66 for the ML classification (see accuracy assessment in S1 Table). The classification was later supplemented with a home class, where homes were randomly assigned in each study location in proportion to the population, with a fixed population size of 200 farmers for each simulation.

*Climate* - We used monthly precipitation (mm) from the climate research unit dataset [50] downscaled to a resolution of 1km$^2$, using the Delta method [48,51]. For each one of the locations modelled, we extracted the raster values and used them in our model as integer values for each month. In addition, we estimated the number of non-rainy days per month from past literature [52].

**Human agent characteristics.** *Farmer activity* - The characteristics and behavior of farmers in the study area (see above) was first characterized via a community survey conducted during two weeks in July 2019. We visited four different divisions in the district of Ratnapura, and in each one we interviewed 10 farmers (40 in total) of different crops: with 22 engaged in rice farming, 22 in tea farming, and 10 in rubber (some farmers tend multiple crops). Each farmer was asked to answer a set of questions related to work schedules, including: planting season, harvest season, hour of starting work, hour of finishing work, seasonal rotation of

**Table 1. A complete list of parameters used in the model for all agent types.** Each of the parameters is either an input for the snake behavior submodel, farmer behavior submodel, or a global variable (climate and landcover).

| Model | Parameter | Value | Source |
|---|---|---|---|
| Farmers | Farmer type | Rice, Rubber, Tea | Field work |
| Farmers | Land type work index | 0–110 | Field work |
| Farmers | Starting hour | 4-9AM | Field work |
| Farmers | Number of hours worked | 4–14 | Field work |
| Farmers | Percentage of population working as farmers | 30–70% | Government reports [53] |
| Snake | Point process models | $0–3*10^{-8}$ | Calculated from snake data (S1 Data) |
| Snake | Seasonal activity probability | 0–1 | Literature [54–56] |
| Snake | Daily activity patterns | 0-.5 | Literature [57] |
| Snake | Aggressiveness | 1–10 | Local herpetologists' questionnaire |
| Snake | Land association factor | 0–2.429 | Calculated from snake data (S2 Table) |
| Land cover | Type of land cover | Rice, Tea, Rubber, Forest, Water, Home | Remote sensing |
| Climate | Mean monthly precipitation | 21–1054 | Climate Research Unit |
| Climate | Number of rainy days | 10–25 | Literature [52] |

crops, as well as size of plot. We also asked farmers about previous encounters with snakes, including location, and season when snakes were encountered. Our final farming dataset included a list of parameters that defined the farming behavior in the model (see Table 1).

Based on the results of the survey, we allowed farmer agents in the model to have the option of moving among up to three different landcover types, and to choose between different working schedules on each landcover type. To take into account the seasonal variation of labour requirements according to the various cropping cycles, we first developed a labor index:

$$I_{ij} = \frac{(247 \times F_{ij}) \div A_i}{30 \div D_{ij}} \tag{1}$$

where $I_{ij}$ is the labor index for landcover $i$ during month $j$ for 1 square kilometer of that landcover, $F_{ij}$ is the number of farmers needed at landcover $i$ during month $j$ for the size of landcover owned by a specific farmer, $A_i$ is the size of landcover $i$ in acres, and $D_{ij}$ is the number of days per month that land cover $i$ is farmed during month $j$, and 247 is used to convert acres (the measurement farmers used when answering the questionnaire) into square kilometers.

A mean value of $I_{ij}$ was calculated using the different index values obtained by the farmers and was distributed between the months according to the working schedule described by the farmers in the interviews. For the rubber landcover the index was calculated for a single day, and then multiplied by the estimated number of non-rainy days that occur in that specific month, since rubber farmers cannot work in the rain due to technical limitations of rubber harvesting methods.

In the model, the probability of each farmer attending each landcover type is then calculated at the beginning of each month:

$$W_{ij} = \frac{S_j \times I_{ij}}{W_{\max}} \times P \tag{2}$$

where $W_{ij}$ is the number of farmers that are going to work in month $i$ in landcover type $j$, $S_j$ is the size of landcover type $j$ in a simulation, $I_{ij}$ is the labor index for month $i$ and landcover $j$ (from Eq 1), $W_{max}$ is the maximum value of $W$ possible for the location being simulated, and $P$ is the farmer population size of the location being simulated. Once a farmer is assigned a certain landcover for month $i$, they will only work on that specific landcover during that month.

The farmers are then assigned a random number from a uniform distribution composed of the possible number of hours farmers work in the field for that specific landcover, based on what was reported by the farmers interviewed during the field work (S3 Table). For the starting hour, the farmers choose a random value out of a normal distribution composed of the possible starting hours for that specific landcover, based on what was reported by the farmers during the field work (S4 Table).

**Snake agent characteristics.**   *Distribution and abundance* - We used Poisson Point Process Models (PPMs) to represent potential abundance of snakes for each species. We interpreted these models as representing the relative carrying capacity and a proxy for potential abundance for each species in each one of the locations modelled in our simulation. In order to calibrate our model's snake population size, we used previous research in which the species *Hypnale hypnale* was systematically surveyed to estimate the number of individuals per square kilometer of forest habitat [58]. This provided a link between PPM outputs and measured snake abundance in forest landscapes, which we then applied to other species and habitat types according to relative model weights following a habitat preference analysis (see below). This method resulted in abundance estimates up to 900 individuals per species per 2x2km tile (= up to 225 snakes per species per km$^2$).

*Habitat preferences* - Preference of landscape for each snake species was defined by a land association factor, calculated using the data points that were used to create the species distribution models. Using chi-square tests, the likelihood of a snake species being found on a specific land cover versus the probability that it would be found there at random was calculated (see S2 Table).

*Activity and behavior* - We incorporated several different measures of snake activity and behavior into the model, including seasonal activity patterns, daily behavioural habits, movement preferences among available habitats, and aggressiveness.

In the model, we assumed that there are a fixed number of snakes for each species present on a tile based on the PPM maps and population size estimate. Changes in activity levels throughout the year were defined according to observed seasonal activity in the tropics [54–56], and according to observations made on *Hypnale spp* [58]. At each monthly update a certain percentage of the snakes from each species becomes active according to the level of precipitation measured (see section 4), as calculated by:

$$A_i = \frac{P_i}{P_{max}} \qquad (3)$$

where $A_i$ is the activity factor for month $i$, and $P_i$ is the precipitation level for month $i$, and $P_{max}$ is the max level of precipitation for the region.

The snake daily activity is determined probabilistically according to the snake activity patterns, with each species being pre-defined as either diurnal, nocturnal, crepuscular, or cathemeral [57]. A probability distribution was designed for each of the different daily activity patterns by identifying hours of sunrise and sunset, and setting the distributions in relation to those hours. All snakes were defined to have a baseline probability of 0.1 (10% chance) for being active even in hours when they are biologically defined as inactive, e.g. nocturnal snakes during daytime, in order to capture the full scope of encounter probability as described by epidemiological surveys (see below).

The probability of snakes moving to a specific landcover type is calculated using the amount of landcover type available and the attraction of the snake to that specific landcover type (see S2 Table for the land association factor). The probability of each species moving to any type of

landcover type was defined by a transition rule as:

$$M_{ij} = \frac{P_j L_{ij}}{P_1 L_{i,1} + \cdots + P_n L_{i,n}}$$ (4)

where $M_{ij}$ is the probability of an individual of snake species $i$ to move to land cover type $j$, $P_j$ is the number of cells of land cover $j$, and $L_{ij}$ is the landcover association factor between snake species $i$ and landcover $j$. After calculating the transition rule, a random number is drawn to decide what landcover the snake will move to.

**Snakebites.** Agents are tracked within the model locations and their encounters (occurring in the same grid cell at the same time) are recorded as snakebites under the following conditions. The probability of a snakebite occurring during an encounter is calculated by taking into account the varying propensities of each species to attack during an encounter. We incorporated aggressiveness by way of an aggressiveness index, which is a ranking of between 1–10 (1 = docile, 10 = very aggressive) as determined by local herpetologists (Table 2). The probability of a snakebite occurring is therefore calculated as:

$$P_i = \frac{A_i}{A_{\max}}$$ (5)

where $P_i$ is the probability of snake species $i$ causing a snakebite when there is a human-snake interaction, $A_i$ is the aggressiveness index for snake $i$, and $A_{max}$ is the maximal value for aggressiveness. When humans and snakes meet on the same cell, a random number is drawn between 0–1, and if it is smaller than the value obtained from the calculation then a snakebite occurs. This function is designed in order to assign a threshold for bites according to each snakes aggressiveness level, with the assumption that combining the aggressiveness along with the human-snake overlap would provide a good measure for snakebites occurring.

**Model evaluation.** We evaluated our model in two different ways: hypothesis testing (verification) and validation. For validation we used the "multiple patterns" methodology in order to check for consistency between the model and the observed data. This was done to make sure we were not overfitting the model, and to make sure it represented the general dynamics of the system [43,59]. For the hypothesis testing we examined the process representation to make sure our model represented both the micro and macro level phenomena correctly, and that the system properly represented the dynamics and mechanism(s) that it is supposed to be representing. For validation we used the model formulations that were chosen during model selection. In addition, for the variables that were tested during the sensitivity analysis we chose variable values that were parameterized using the analysis output in order to make sure the

**Table 2. Snake behavior profiles for each species, as reported by local expert herpetologists.** These profiles were integrated into the snake agent behavior variables, with the aggressiveness index and dial activity directly integrated into the model, and zonation is given as a broad description while the habitat preference factor was used in order to define snake behavior.

| Species | Common name | Aggressiveness | Daily activity | Zonation |
|---|---|---|---|---|
| *Daboia russelli* | Russell's viper | 8 | Nocturnal | Terrestrial |
| *Echis carinatus* | Saw scaled viper | 10 | Cathemeral | Terrestrial |
| *Hypnale hypnale* | Hump nosed viper | 10 | Nocturnal | Semi-arboreal |
| *Hypnale zara* | Hump nosed viper | 10 | Nocturnal | Semi-arboreal |
| *Hypnale napa* | Hump nosed viper | 10 | Nocturnal | Semi-arboreal |
| *Bungarus caeruleus* | Common krait | 2 | Nocturnal | Terrestrial |
| *Bungarus ceylonicus* | Ceylon Krait | 1 | Nocturnal | Terrestrial |
| *Naja naja* | Cobra | 5 | Cathemeral/ Crepuscular | Semi-aquatic |

values were above a threshold that allowed emergent patterns to appear in our system. For the full description of model selection and sensitivity analysis see S1 Appendix.

**Validation.** For external validation we chose multiple patterns on which there was already research conducted in Sri Lanka, such as temporal patterns of snakebites [7], the relative risk of snakebite between locations [12], and biting snake species composition among bite victims as inferred from hospital records [37]. This was done in accordance with the POM protocol [43], which suggests that multiple patterns be assessed and the fit between the model predictions and these patterns evaluated (as opposed to comparing results to a single statistic or a single pattern). This is supposed to prevent overfitting of the model to an expected output, or falsely representing the model by using only one output parameter, and to make sure that the model can represent the dynamics of the system that it is attempting to represent.

## Hypothesis testing

We checked for consistency of process representation, following the spatial and temporal patterns of the snake and farmer agents, and snakebites. We did this for the distribution of snakebites across both the months of the year and across the hours of the day. We then checked when peak snakebites were occurring and their relationship to the movement patterns of the agents. This allowed us to make sure that the system was properly representing both the micro level (agents' movements) and the macro level (snakebite distribution) and the relationship between them.

## Hypothesis generation

The POM protocol also suggests looking for secondary predictions that emerge from the model and using them later for further validation if observations become available, and if not then using them to prompt further research in the field [42]. We checked for the following secondary predictions: monthly and daily patterns by snake species, by division, and by landcover types.

## Results

### Validation

Overall, the model performed well in differentiating between high and low risk locations. The results are based on simulation runs for 45 different locations across the entire district of Ratnapura, with high and low defined as above or below the median snakebite risk for all locations simulated. Predictions of the ABM showed a significant difference in prediction between locations where snakebite risk was above the median of all locations simulated and those where snakebite risk was below the median using Welch two sample t-test (t = -5.539, df = 39.20, p-value < 0.001) (S2 Fig).

The model also effectively predicted the relative contribution of different species to overall snakebite patterns as derived from hospital surveys [37], both in divisions 1–3 (Eheliyagoda, Balangoda, and Kalawana) which were located in the wet zone, and divisions 4 (Emptilipitiya) which was located in the intermediate zone (Table 3 and Fig 4). The simulation contribution of cobras was overestimated in our model in all locations, and the contribution of Russell's viper and hump nosed viper against entire snakebites were underestimated in the intermediate zone. Additionally, in contrast to the hospital survey our model did not include non-venomous species, so an over estimation is to be expected to a certain extent.

The model was also successful in predicting the temporal patterns of snakebite in Sri Lanka reported previously. Snakebite has been reported as having three peaks in general throughout

**Table 3. The average predicted proportion of bites from different snake species across four different locations.** The first three divisions (Balangdoa, Eheliyagoda, Kalawana) belong to the wet zone of Sri Lanka, while the fourth region (Embilipitiya) belongs to the intermediate zone of Sri Lanka.

| Wet zone (1–3) | Model prediction | Hospital data |
|---|---|---|
| Hump nosed viper | 51–57% | 65% |
| Russell's viper | 21–24% | 25% |
| Cobra | 23–26% | 5% |
| Non-venomous species | | 5% |
| Intermediate zone (4) | Model prediction | Hospital data |
| Russell's viper | 39% | 50% |
| Hump nosed viper | 16% | 30% |
| Common Krait | 10% | 10% |
| Cobra | 33% | 5% |
| Non-venomous species | | 5% |

the year (November–December, March–May, August), although there are regional variations [7]. The ABM predicted the possibility of different main peaks of snakebites through the year, including a large peak in March-May (Balangoda, Eyeliyagoda, Kalawana, Embilipitiya), a second peak around August (Balangoda, Kalawana), and a third peak in November-December (Balangoda, Eyeliyagoda, Kalawana, Embilipitiya) (Fig 5).

## Hypothesis testing

The model performed well in representing the micro level (agent movement) and its relation to the macro level (snakebite distribution), with a clear pattern of spatial-temporal overlap between snakes and farmers as the cause of snakebites (Fig 6). The highest frequency of snakebite during the year occurred when both farmers and snakes were present and active on the different landcover types, although bite frequency differed among landcover types. On tea plantations, snakebites are simulated to follow snake activity closely as the activity level of farmers is highly consistent throughout the year (Fig 6A–6D). Since the level of snake activity is defined by the amount of precipitation, the snakebites patterns follow seasonal rainfall

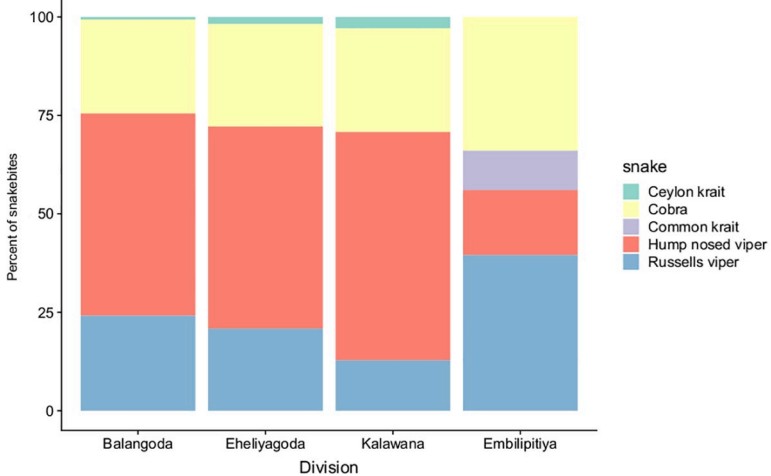

**Fig 4. The average predicted proportion of bites from different snake species across four different locations.** The first three divisions (Balangdoa, Eheliyagoda, Kalawana) belong to the wet zone of Sri Lanka, while the fourth region (Embilipitiya) belongs to the intermediate zone of Sri Lanka.

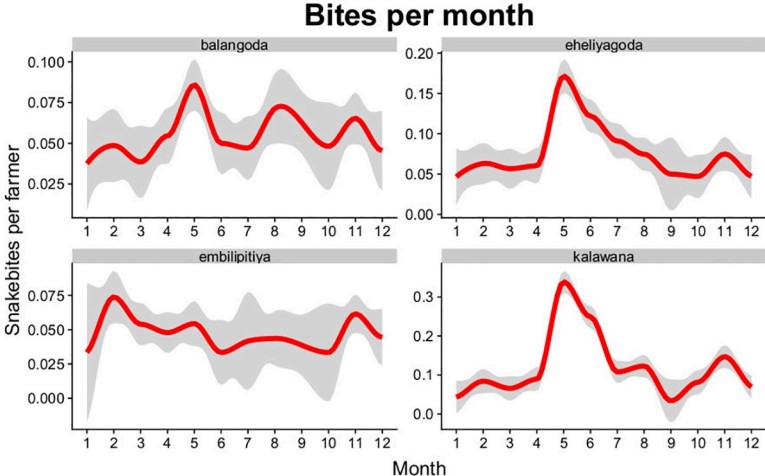

**Fig 5. Snakebites per farmer across different months.** Results are based on 30 simulation runs for each location across 4 divisions representing snakebite patterns across the year.

(Fig 6A–6D). For rice paddies, snakebite peaks occur at different time periods–either in April-May (peak snake activity), in August (peak farmer activity), or November (a combination of both) (Fig 6B–6E). This reflects seasonal variability of rice farmers' behaviors, which have a different activity peak from snakes (Fig 6B–6E). On rubber plantations, snakebites are a mixture of both snake and farmer activity as well, with the highest peak in bites occurring when snakes are most active in April-June (Fig 6C–6F).

Distinct patterns of spatio-temporal overlaps on the daily level are also evident. For the tea landcover, peak activity tends to follow a bimodal pattern with peaks occurring in both late afternoon and early morning (S3A Fig). This pattern reflects the working pattern of tea farmers that tend to start working early during the day, but also follow long working hours, which results in farmers meeting snakes both when snakes are active early morning, and when snakes are active during late afternoon. For the rice land cover, snakebites have the highest probability of occurring during late afternoon when farmers and snakes have high overlap, but may also occur in the early morning during peak activity months (S3B Fig). This pattern reflects the

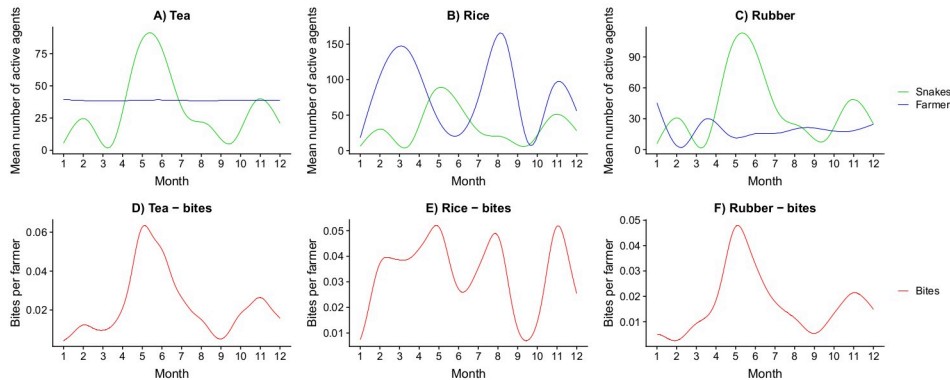

**Fig 6. Spatio-temporal overlap between farmers and snakes for each land cover type.** Values represent the mean number of farmers, snakes, and bites for 660 simulation runs across all locations. Each graph in the first row follows the monthly spatio-temporal overlap between farmers and snakes for **A)** tea **B)** rice, and **C)** rubber, and each graph in the second row follows the snakebite pattern that emerges out of the spatio-temporal overlaps for **D)** tea **E)** rice, and **F)** rubber.

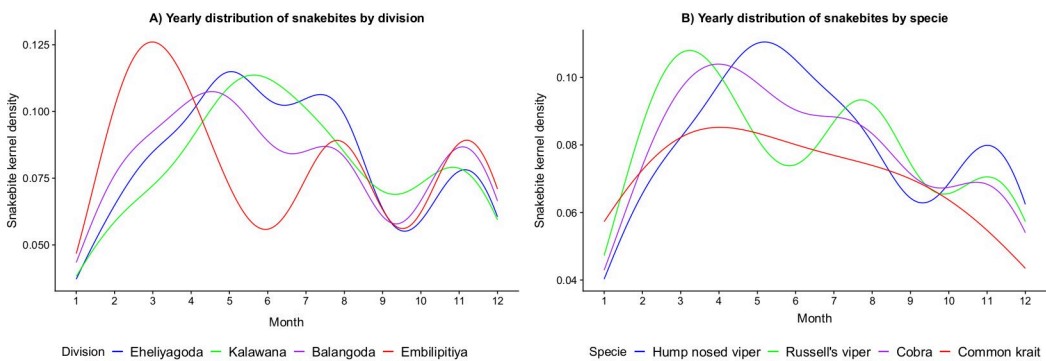

**Fig 7. Secondary predictions A) the yearly distribution of snakebites for different divisions**. Each division showed a distinct pattern of snakebite, with the largest peak of the year varying between March and May. **B) The yearly distribution of snakebites for different species**. Each species showed different snakebite peaks through the year, with the largest peak occurring between February and May.

working pattern of rice farmers that tend to start later during the day, but work for long hours, there for increasing the chances of encounter while snakes are active later in the day. For rubber, snakebites have the highest probability of occurring during the early hours of the morning (S3C Fig). This pattern reflects the working pattern of rubber farmers that tend to start working early in the day when snakes are active, but also have short working hours, so a second snakebite peak later in the day does not occur.

## Hypothesis generation

A secondary prediction of our model was that the monthly burden of snakebites varies across locations, (Fig 7). Our model predicted that in drier locations the peak in bites occurs earlier in the year during February-April, whereas wetter locations tend to have a higher peak in bites during the month of May (Fig 7). The different patterns cannot be traced to a single factor but is likely caused by a combined effect of land cover and climatic differences, and the interaction between snakes, farmers, and their environment within these locations (see S4, S5, S6, S7 and S8 Figs). This prediction also suggests that there may be significant temporal differences in snakebites between the wet, dry, and intermediate zones in Sri Lanka.

Another secondary prediction from our model estimates that the monthly distribution of snakebites varied between species, with a different pattern for each species (Fig 7). These different patterns are not caused by snake activity alone, but by a combination of snake habitat preference, snake activity, and the seasonal patterns of farmers on different landcover types.

## Discussion

Snakebite affects poor and rural populations that are exposed to venomous snakes, yet few studies have attempted to decompose spatial and temporal patterns and predict risk on the basis of social-ecological causative mechanisms. Here we develop a mechanistic model to examine snakebite dynamics by simulating snake-human encounters in rural agricultural communities using an agent-based model (ABM). Our simulation represents the farmer-snake interactions that are driving snakebite patterns in Sri Lanka, a bite hotspot country within the highly affected South Asian region. While it has been previously shown that snakebites can have strong spatial and temporal patterns [12,37], and different studies have explored these patterns on local scales [60,61], our model provides a unique mechanistic perspective regarding the emergence of these patterns from basic ecological principals regarding species

interactions on a more local scale. Results showed that the model performed well in simulating snakebite occurrences across spatial and temporal scales, including daily and seasonal patterns, biting species assemblages, and bite incidence variation among locations (Figs 4,5, 6 and S2–S8).

The results suggest that the risks of snakebite depend on factors influencing the behaviors of both farmers and snakes, including landcover, precipitation, and the interaction between humans and snakes (Figs 6 and 7). Our model also concurs with previous research showing that seasonal precipitation patterns dictate patterns of snakebites by influencing the activities of both snakes and farmers (Fig 6) [4,12]. We further discovered that different crop types result in distinct work schedule in relation to daily human activities and rainy seasons, greatly altering overall risk profiles of snakebites for each crop (Fig 6E, 6F and 6G). Additionally, the composition of snake species is different among various crop types (S8 Fig), leading to complex social-ecological interactions that in turn contribute to snakebite risk [14].

Our model suggests greater resolution on the composition of species delivering bites is essential in order to better resolve snakebite risks in future (Fig 4). Previous research has supported the idea that following the ecology and behavior of each species would give a better understanding of both the mechanism driving bite patterns for individual snake species [18], and for different types of landcover (e.g.: [62]). Our model provides a mechanistic explanation for the ways snake ecology and human behavior combine to result in species specific snakebite patterns. For example, in our study system, although two species (Russell's vipers and Hump nosed vipers) show similar seasonal activity patterns, a stronger preference for rice paddies for one of the species (Russell's vipers) and a stronger preference for rubber plantations in the other species (Hump nosed vipers) results in very different temporal patterns of encounter. Understanding the overall pattern of snakebite therefore requires understanding of the specific ecology of each species (Fig 7B).

Such differences in an example of why predicted snakebite patterns vary considerably between locations, since spatial heterogeneity of famer types and snake species create fine scale differences in encounter risk, a prediction which concurs with previous research [12,13,37,63]. In our study, this difference between locations was in practice caused by a combination of factors, including different distributions of key landcover types and climatic conditions, which in turn affect either snakes or famers or both. For example, the division of Embilipitiya, which is located in the intermediate climatic zone of Sri Lanka, had a less suitable environment for Hump-nosed vipers and a high concentration of rice paddies, resulting in a snakebite pattern different, including overall risk, temporal patterns of risk and biting species composition, to those found in the sites in the wet zones (Figs 4, 5 and 7).

Our study clearly showed that the spatio-temporal synchronicity in both snake and farmer behaviors is the key to understand the snakebite patterns in the Ratnapura district in Sri Lanka (Figs 6 and S3). In particular, multiple climatic profiles within the district may result complex snake-farmer associations evident from the snakebites patterns as well as the composition of responsible snake species (Figs 4, 5 and 7). While our study shows a strong implication of social-ecological dynamisms of snakebites in dry and wet-dry climate zones in Sri Lanka, other studies have already invoked similar mechanisms to explain observed patterns of risk in rural communities outside of Sri Lanka (e.g.: [14,17]). Considering the ease of re-parameterizing simulation models to generate baseline snakebite risk predictions on any spatial and temporal scale, our model has strong potential for applications in other areas across the tropics. For example, locations outside of Sri Lanka that include some of the same venomous snake species have shown yearly temporal distributions of snakebites that contrast with those observed inside of Sri Lanka [16,17,64], which provides a strong avenue for hypothesis generation and testing of the model in different systems. Outside of Sri Lanka, other studies have similarly

reported land-use specific risks (e.g. rubber in Liberia and rice in the Philippines) [65,66]. Transferring the model to these regions could shed further light on the combinations of factors that underpin different snakebite patterns among different locations, again a potentially fruitful avenue for hypothesis generation or validation.

While our model represented some of the most important snake behavior factors relevant to snakebite, there are other elements that we did not address, primarily due to data limitations. These include reproduction phenology and its association with climate [4], seasonal variability in landcover preferences [58], or feeding habits and species-specific feeding strategies. For example, it is known that reproductive behavior can increase the chances of encountering snakes [67,68], and integrating this behavior into the model may improve predictions. Additionally, differences between feeding strategies such as active hunting (e.g. *Naja naja*) and ambush (e.g. *Hypnale hypnale* & *Daboia russellii*) may lead to different encounter outcomes, and integrating these traits may reduce the overestimation of cobra bites in comparison to other snake species and improve our predictions for the Ratnapura district. Similarly, we have not captured all the behavioral traits of farmers, such as differences in farming practices between small and large plantations, seasonal crop rotations [69], and additional crop types (e.g., small gardens, cinnamon, banana, coconut) [45], adaptive characteristics that represent farmers' planning strategies over multiple years, or specific behaviors relating to snakebite epidemiology, such as health seeking behavior or the use of protective measures (e.g., boots) [70]. Additionally, we did not integrate the distance between homes and fields due to limitations of our modeling framework, even though it has been known to be an important factor for snakebite occurrence. Nevertheless, our model has demonstrated the importance of integrating both human and snake behavior into a single model and has shown that integrating even a few essential characteristics can have strong explanatory value for predicting patterns of snakebite.

Snakebite is an ongoing concern in Sri Lanka, and across southern Asia and much of the tropical and subtropical developing world. The World Health Organization has launched a strategic plan to reduce snakebite injuries and mortality by 50% by the year 2030, yet it has been suggested that one of the key barriers to preventing snakebite is the lack of good quality research to help direct effort [36]. Here we explored fine scale spatially explicit predictions by developing a novel mechanistic model to explain snakebite risks based on snake behaviors (e.g. snake activities and distributions) and farmer behaviors (e.g. work schedules for different landcover types). Our approach is based on clear, general mechanisms and strong socio-ecological theory and is therefore highly transferrable to other systems, where the risks of snakebite are similarly associated with occupational characteristics, environmental conditions and snake ecological traits [8,17,19,71–73]. Our model, once implemented with local datasets, can examine the local socio-ecological drivers of snakebites and predict spatial and temporal snakebite patterns, as well as generating hypotheses and testing the efficacy of policy intervention. With snakebite burden in Sri Lanka expected to increase under climate change [7] our findings carry important implications for future snakebite prevention in the study sites where it was developed. The insights gained in this study will help to focus future efforts to collect relevant data and resolve key mechanisms underlying snakebite risk, which should help advance management planning and the direction of scarce management resources.

## Supporting information

**S1 Fig. Model outline.** A. model outlines B. model structure.
(DOCX)

**S2 Fig. Model output of mean snakebite risk for locations with high and low snakebite occurrence.**
(DOCX)

**S3 Fig. The daily spatial temporal overlap of farmers and snakes for rice farmers.** A. Rice farmers B. Tea farmers C. Rubber farmers.
(DOCX)

**S4 Fig. Daily distribution of snakebite by division.**
(DOCX)

**S5 Fig. Daily distribution of snakebite by landcover type.**
(DOCX)

**S6 Fig. Daily distribution of snakebite by snake specie.**
(DOCX)

**S7 Fig. Yearly distribution of snakebite by landcover type.**
(DOCX)

**S8 Fig. Percent of snakebite by snake specie.**
(DOCX)

**S1 Table. Classification assessment.** A. Support vector machine B. Maximum likelihood.
(DOCX)

**S2 Table. Land association factor.**
(DOCX)

**S3 Table. Farmers working hours.**
(DOCX)

**S4 Table. Farmers starting hours.**
(DOCX)

**S1 Data. PPM at different locations.**
(XLSX)

**S1 Appendix. Technical evaluation.** A. Model selection B. Sensitivity analysis C. Results.
(DOCX)

## Acknowledgments

We are grateful to Ruchira Somaweera for curation of snake occurance data, and to Udaya Wimalasiri for assistance during the field work.

## Author Contributions

**Conceptualization:** Eyal Goldstein, Kris A. Murray, Takuya Iwamura.

**Data curation:** Eyal Goldstein, Joseph J. Erinjery, Gerardo Martin, Anuradhani Kasturiratne, Dileepa Senajith Ediriweera, Hithanadura Janaka de Silva, Takuya Iwamura.

**Formal analysis:** Eyal Goldstein, Takuya Iwamura.

**Funding acquisition:** Hithanadura Janaka de Silva, Peter Diggle, David Griffith Lalloo, Kris A. Murray, Takuya Iwamura.

**Investigation:** Eyal Goldstein, Joseph J. Erinjery, Gerardo Martin, Kris A. Murray, Takuya Iwamura.

**Methodology:** Eyal Goldstein, Kris A. Murray, Takuya Iwamura.

**Project administration:** David Griffith Lalloo, Kris A. Murray, Takuya Iwamura.

**Software:** Eyal Goldstein, Takuya Iwamura.

**Supervision:** Joseph J. Erinjery, Kris A. Murray, Takuya Iwamura.

**Validation:** Eyal Goldstein, Joseph J. Erinjery, Takuya Iwamura.

**Writing – original draft:** Eyal Goldstein, Takuya Iwamura.

**Writing – review & editing:** Joseph J. Erinjery, Gerardo Martin, Anuradhani Kasturiratne, Dileepa Senajith Ediriweera, Hithanadura Janaka de Silva, Peter Diggle, David Griffith Lalloo, Kris A. Murray, Takuya Iwamura.

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
