## [Decision Letter · Decision Letter 0]

27 Oct 2020

Dear Dr Goldstein,

Thank you very much for submitting your manuscript "Integrating human behavior and snake ecology with agent-based models to predict snakebite in high risk landscapes" for consideration at PLOS Neglected Tropical Diseases. As with all papers reviewed by the journal, your manuscript was reviewed by members of the editorial board and by several independent reviewers. The reviewers appreciated the attention to an important topic. Based on the reviews, we are likely to accept this manuscript for publication, providing that you modify the manuscript according to the review recommendations. 

Please do find the two attachments accompanying the reviews..

Sincerely,

Abdulrazaq G. Habib

Guest Editor

José Gutiérrez

Deputy Editor

Please do find the two attachments accompanying the reviews..

Reviewer's Responses to Questions

**Key Review Criteria Required for Acceptance?**

**Methods**

-Are the objectives of the study clearly articulated with a clear testable hypothesis stated?

-Is the study design appropriate to address the stated objectives?

-Is the population clearly described and appropriate for the hypothesis being tested?

-Is the sample size sufficient to ensure adequate power to address the hypothesis being tested?

-Were correct statistical analysis used to support conclusions?

-Are there concerns about ethical or regulatory requirements being met?

Reviewer #1: The method is relevant and well explained.

Reviewer #2: (No Response)

Reviewer #3: The introduction is adequate, objectives are well presented.

The study design appropriate

**Results**

-Does the analysis presented match the analysis plan?

-Are the results clearly and completely presented?

-Are the figures (Tables, Images) of sufficient quality for clarity?

Reviewer #1: The results are presented according to the analysis plan and clearly exposed.

Reviewer #2: (No Response)

Reviewer #3: The data generated are well articulated

**Conclusions**

-Are the conclusions supported by the data presented?

-Are the limitations of analysis clearly described?

-Do the authors discuss how these data can be helpful to advance our understanding of the topic under study?

-Is public health relevance addressed?

Reviewer #1: The limitations of the model are mentioned.

The authors clearly show implication of the model and its public health relevance, in Sri Lanka but also in other regions of the world where snakebite envenomation is an issue.

The conclusions are consistent and reliable.

Reviewer #2: (No Response)

Reviewer #3: The conclusion need editing to relate the finding of the study to real time realities in the study sites

**Editorial and Data Presentation Modifications?**

Reviewer #1: Minor comments:

Page 17, 2nd paragraph, line 3: divisions 4 (not divions 4)

Reviewer #2: (No Response)

Reviewer #3: (No Response)

**Summary and General Comments**

Reviewer #1: Goldstein et al’s manuscript “Integrating human behavior and snake ecology with agent-based models to predict snakebite in high risk landscapes” described an agent based modelling to explain the determinants of man – snake encounter. Results of the study corroborate the observations from several authors and are both convincing and rational.

Although it has been hypothesized since a long time that human-snake encounter was influenced by combinations between human activity and snake behavior (in addition to references quoted by the authors: Chippaux et al. Bull Soc Path Exot. 1981; 74 (4): 458-67; Sawai et al. Problem snake management. The Habu and the Brown Treesnake. Cornell University Press, Ithaka, 1999:107-15; Bochner & Struchiner. Cad Saude Publica. 2004;20(4):976-85; Chippaux. Snake venoms, Envenomations. Krieger Publishing Co, Malabar, 2006; Stock et al. Nat Biotechnol. 2007; 25 (2): 173-7; Chippaux. Med Sci (Paris). 2009;25(10):858-62), it had never been proposed a model –particularly an agent based modeling– to predict its impact and define a preventive strategy.

I am not an expert in modeling and I cannot give a relevant opinion on this topic. However, on the one hand the approach is perfectly appropriate and justified and, on the other hand, the variables considered to build the model seem adequate.

As pointed by the authors, some determinants concerning snake population and behavior should be considered in more sophisticated models. For example, several studies have shown that during the mating period, males were encountered 5 to 10 times more than in other seasons of the year and, at the same moment, than females (see Wang et al. Zool Stud. 2003;42(2):379-85; Chippaux. Med Sci (Paris). 2009;25(10):858-62; Bauwens & Claus. Ecol Evol. 2019;9(10):5821-34), which proportionately increases the risk of snakebites. However, it is not certain that such improvements lead to significantly better predictions.

For more detailed analysis of the relationship between human activities and snake behavior, the authors should refer to the chapter by Sawai et al. (in Problem snake management. The Habu and the Brown Treesnake. Cornell University Press, Ithaka, 1999:107-15).

Reviewer #2: (No Response)

Reviewer #3: The study is novel and well presented

PLOS authors have the option to publish the peer review history of their article (what does this mean?). If published, this will include your full peer review and any attached files.

Reviewer #1: No

Reviewer #2: Yes: Nafiu Hussaini, PhD

Reviewer #3: No
---

## [Decision Letter · Decision Letter 1]

7 Dec 2020

Dear Dr Goldstein,

We are pleased to inform you that your manuscript 'Integrating human behavior and snake ecology with agent-based models to predict snakebite in high risk landscapes' has been provisionally accepted for publication in PLOS Neglected Tropical Diseases.

Best regards,

Abdulrazaq G. Habib

Guest Editor

José Gutiérrez

Deputy Editor

Reviewer's Responses to Questions

**Key Review Criteria Required for Acceptance?**

**Methods**

-Are the objectives of the study clearly articulated with a clear testable hypothesis stated?

-Is the study design appropriate to address the stated objectives?

-Is the population clearly described and appropriate for the hypothesis being tested?

-Is the sample size sufficient to ensure adequate power to address the hypothesis being tested?

-Were correct statistical analysis used to support conclusions?

-Are there concerns about ethical or regulatory requirements being met?

Reviewer #2: (No Response)

Reviewer #3: (No Response)

**Results**

-Does the analysis presented match the analysis plan?

-Are the results clearly and completely presented?

-Are the figures (Tables, Images) of sufficient quality for clarity?

Reviewer #2: (No Response)

Reviewer #3: (No Response)

**Conclusions**

-Are the conclusions supported by the data presented?

-Are the limitations of analysis clearly described?

-Do the authors discuss how these data can be helpful to advance our understanding of the topic under study?

-Is public health relevance addressed?

Reviewer #2: (No Response)

Reviewer #3: (No Response)

**Editorial and Data Presentation Modifications?**

Reviewer #2: (No Response)

Reviewer #3: (No Response)

**Summary and General Comments**

Reviewer #2: (No Response)

Reviewer #3: (No Response)

PLOS authors have the option to publish the peer review history of their article (what does this mean?). If published, this will include your full peer review and any attached files.

Reviewer #2: No

Reviewer #3: No

---

## [Editor Report · Acceptance letter]

17 Jan 2021

Dear Mr. Goldstein,

We are delighted to inform you that your manuscript, "Integrating human behavior and snake ecology with agent-based models to predict snakebite in high risk landscapes," has been formally accepted for publication in PLOS Neglected Tropical Diseases.

Best regards,

Shaden Kamhawi

co-Editor-in-Chief

Paul Brindley

co-Editor-in-Chief
